# GEM-based computational modeling for exploring metabolic interactions in a microbial community

**Soraya Mirzaei**[1], **Mojtaba Tefagh**[1,2]*

**1** Department of Mathematical Sciences, Sharif University of Technology, Tehran, Iran, **2** Center for Information Systems & Data Science, Institute for Convergence Science & Technology, Sharif University of Technology, Tehran, Iran

* mtefagh@sharif.edu

## Abstract

Microbial communities play fundamental roles in every complex ecosystem, such as soil, sea and the human body. The stability and diversity of the microbial community depend precisely on the composition of the microbiota. Any change in the composition of these communities affects microbial functions. An important goal of studying the interactions between species is to understand the behavior of microbes and their responses to perturbations. These interactions among species are mediated by the exchange of metabolites within microbial communities. We developed a computational model for the microbial community that has a separate compartment for exchanging metabolites. This model can predict possible metabolites that cause competition, commensalism, and mutual interactions between species within a microbial community. Our constraint-based community metabolic modeling approach provides insights to elucidate the pattern of metabolic interactions for each common metabolite between two microbes. To validate our approach, we used a toy model and a syntrophic co-culture of *Desulfovibrio vulgaris* and *Methanococcus maripaludis*, as well as another in co-culture between *Geobacter sulfurreducens* and *Rhodoferax ferrireducens*. For a more general evaluation, we applied our algorithm to the honeybee gut microbiome, composed of seven species, and the epiphyte strain *Pantoea eucalypti 299R*. The epiphyte strain Pe299R has been previously studied and cultured with six different phyllosphere bacteria. Our algorithm successfully predicts metabolites, which imply mutualistic, competitive, or commensal interactions. In contrast to OptCom, MRO, and MICOM algorithms, our COMMA algorithm shows that the potential for competitive interactions between an epiphytic species and Pe299R is not significant. These results are consistent with the experimental measurements of population density and reproductive success of the Pe299R strain.

## Author summary

Microbial consortia play critical roles in human health and environmental biogeochemical cycles. Studying interactions and communications among organisms is integral for understanding the role of individual microbes in microbial communities. Organisms release

**Data Availability Statement:** All the information and implementation of algorithms can be found in the GitHub repository: https://github.com/mirzaei-s/microbial_interactions. This repository contains the implementation of mixed-integer optimization

problem and linear programming for COMMA and CPARMA, related to competition and parasitism/commensal. Additionally, it includes scripts for comparison with OptCom and MICOM.

**Funding:** The author(s) received no specific funding for this work.

**Competing interests:** The authors have declared that no competing interests exist.

and consume metabolites in the extracellular environment, which dictate the assembly and interactions of communities. Researchers have indeed used constraint-based metabolic modeling to analyze the metabolic interactions for multi-organism communities. We propose a computational framework to predict metabolic interactions between pairs of microbial species. We investigate whether, for a specific shared metabolite in the extracellular environment, the two species may have competitive, parasitic, or commensal interactions. These results of our algorithms may help identify important metabolites that shape the interaction patterns in natural communities, and can also be useful for designing media. We demonstrate the applicability of our method by applying it to well-known communities in the honey bee gut microbiota and on leaf surfaces. We then compare the results of our algorithm for the phyllosphere bacterial community with empirical data.

## Introduction

The community of microbes plays a fundamental part in both the environment and human well-being [1–4]. Due to the falling cost of 16S rRNA and shallow shotgun sequencing, we now have access to information about the composition and abundance of microbes in a community [5, 6]. Major shifts in the compositional diversity of these communities are often associated with hampering their functions. This is particularly evident in the case of various diseases, such as type 2 diabetes and obesity, which have been connected to alterations in composition of the gut microbiome [7, 8]. The microbiome community consists of numerous diverse species that interact with each other, resulting in complex and challenging system to understand. The identification of interactions between species is crucial for understanding microbes and their interplay among various microorganisms. Interactions within microbial consortia can have either a positive effect or negative effect. Positive effects can arise from cooperation, commensalism, and mutualism interactions, while negative effects can stem from competition, parasitism, and amensalism. These six categories of microbe-microbe interactions can be primarily attributed to the utilization and sharing of metabolic resources within a microbial community [9–11].

Considering the significance of microbial interactions, various mathematical approaches and tools have been developed [12–17]. One key approach for studying microbial communities *in silico* is stoichiometric constraint-based modeling. In such approaches, *genome-scale metabolic networks* (GEM) are employed to reveal the biosynthetic and metabolic capabilities of all organisms within the community [18]. In an effort to investigate various types of interactions, researchers have constructed synthetic microbial communities by employing GEM models of strains. Additionally, they have devised constrain-based model optimization approaches for optimizing the community-level objective. However, defining an objective function for the metabolism of a microbial community presents a significant challenge. Some GEM-based interaction prediction approaches compare the growth rates computed by single- or bilevel optimization methods [19, 20] in a community with the growth rate of individual microorganisms, while imposing various media conditions. By analyzing the increase or decrease in the growth rate of microbes, predictions about the type of interaction that both microorganisms have with each other were determined [21–25]. In the majority of these methods, the objective function is the sum of biomass fluxes of all species in the community. A common issue encountered is that the optimization problem has a non-unique solution vector, which can result in a non-realistic growth rate for community members [26]. Metagenomic approaches that quantify the relative abundances of taxa in a community were employed to

describe the objective function in GEM modeling, in order to determine the impact of each microbe on interactions within the community [5, 27]. The method of the mixed-integer linear programming algorithm has been used to identify minimal media for a synthetic community of pairs of two species, and then predict the type of interactions by testing whether each species can grow individually on the designed media or not [28]. This strategy is computationally expensive and not appropriate for a microbial community.

Some approaches have been proposed to predict the interactions of species, based on the seed set detection algorithms. The seed set is the minimal number of metabolites that are required for the growth of species in the community. The methods then search for the common metabolites in the seed set, which represent the similarity in two species' nutritional requirements. In some strategies, these seed sets were calculated by using mixed-integer linear programming. As previously stated, this strategy required high computing complexity [29]. An alternative approach is a graph-based algorithm, which has been used to compute the seed sets, and as a result, determine the shared metabolites for which the two species may compete [30–32]. The drawback of these methods is that they did not take into account the steady state and the mass balance. Recently, the combination of graph-based and constraint-based strategies has been proposed to specify how the species in a community influence one another through the metabolic interactions [33].

In genetic engineering of microorganisms with goal of achieving maximum efficiency, it is essential to understand inter-organism interactions and how they compete for resources within the microbial community [34, 35]. The exchange of metabolites plays a crucial role in microbial community assembly. These communities contain thousands of species, many of which cannot be cultured in the laboratory. This necessitates the development of computational models. To date, existing methods face significant challenges, such as defining community-level objective functions and computational complexity, which need to be addressed. In this paper, we develop a method based on the constraint-based modeling approach in systems biology, to investigate metabolic interactions between pairs of microbial species. We investigate, whether for a specific shared metabolite, the two species may have competitive, parasitic or commensal interactions. The key idea is that we explore systematic analyses of the flux distribution space of reactions in a merged-scale metabolic network to determine whether given two microbes have a trade-off between the common substrates. Our computational model does not require prior knowledge of metabolic objective functions of the microbial community.

We applied our framework to two well-studied species pairs interaction: one between *Desulfovibrio vulgaris* and *Methanococcus maripaludis*, and another between *Geobacter sulfurreducens* and *Rhodoferax ferrireducens*. The interaction between *Desulfovibrio vulgaris* and *Methanococcus maripaludis*, which occurs naturally, represents an ecologically relevant microbial mutualism. In this interaction, a hydrogen-producing bacterium (D. vulgaris) interacts with a methanogen archaeon (M. maripaludis) [12]. *G.sulfurreducens* and *R.ferrireducens* indeed exist in aquatic environments, and they exhibit the remarkable ability to perform dissimilatory metal reduction within their anoxic ecological niches [36].

As for the microbial community, we focused on the honeybee gut microbiome and the phyllosphere bacteria community. The honey bee gut microbiota has become an attractive model for investigating fundamental aspects of gut microbiology due to its relatively simple community composition [37]. Microorganisms that colonize the leaves impact the health of plants and humans. Studying the interactions between these microbes is important for gaining insights into bacterial growth and how bacteria interact with each other on the leaf surface (phyllosphere). We also applied our method to obtain insights into seven bacterial interactions that co-colonize non-parasitically in the phyllosphere environment. Our results explain the

changes in population density and cell division of *Pantoea eucalypti 299R* (Pe229R) due to the presence of epiphytic bacterial strains, despite the fact that resource overlap alone cannot account for these changes.

## Materials and methods

### Ethics statement

This research is purely computational and does not involve human subjects or animals. No informed consent or ethical approvals were required. We adhered to best practices for data privacy and followed relevant regulations.

### Species and community metabolic modeling

We obtained genome-scale metabolic networks of phyllosphere bacteria from [38] (https://github.com/roschlec/paper_cusper_competition/tree/main/models). Additionally, we obtained the honey bee community metabolic networks as described by [39] (https://github.com/EPFL-LCSB/remind). The models were reconstructed using an automatic GEM reconstruction tool [40], and they were validated and gap-filled. A GEM network comprises a set of metabolites, denoted as $M = \{M_1, \ldots M_m\}$, where $m$ represents the number of metabolite set. Additionally, $R = \{R_1, \ldots R_n\}$ denotes the set of metabolic reactions, with $n$ representing the size of the reaction set. Furthermore, the stoichiometric matrix $S$, which represents the collection of coefficients in metabolic reactions that occur within the system. Each entry $S_{ij}$ is the stoichiometric coefficient of $i$th metabolite participating in the $j$th reaction. A reaction is considered irreversible if it occurs only in the forward direction, or reversible if the reaction can occur in the reverse direction as well. Reactions that involve the transfer of metabolites between the system and its extracellular environment are considered as exchange reactions.

In our approach, we integrate the individual genome-scale metabolic networks to construct a synthetic community with a stoichiometric matrix $S^c \in \mathbb{R}^{m \times n}$, as described in [28]. From now on, $n$ and $m$ represent the sizes of reaction and metabolite sets in the joint GEM of the pairwise community. Microbes were joined through an extracellular compartment, which served as a representation of the shared compartment allowing metabolite exchange between two species and the environment, as depicted in Fig 1. In the synthetic community, the exchange reactions of shared metabolites in each species' model become internal reactions.

### Toy model

We use toy models of two microbial species, which have two shared metabolites. These models have been published and are publicly available in [41]. All reactions in these models are reversible, allowing both species to produce and secrete the shared metabolites, succinate and ammonium (Fig 2). All the reactions have been converted into two irreversible reactions. In Fig 2, we have depicted these irreversible reactions with double lines.

### D.vulgaris and M.maripaludis models

We applied our algorithm to a syntrophic co-culture of *Desulfovibrio vulgaris* and *Methanococcus maripaludis*. We used the stoichiometric model for this pair, as published in [28] and publicly available in https://synthetic-ecology.bu.edu/models. *Desulfovibrio vulgaris* produces acetate, carbon dioxide, and hydrogen during the fermentation of lactate. *Methanococcus maripaludis* consumes hydrogen, which facilitates the growth of *Desulfovibrio vulgaris*, thus engaging in metabolic mutualistic interactions.

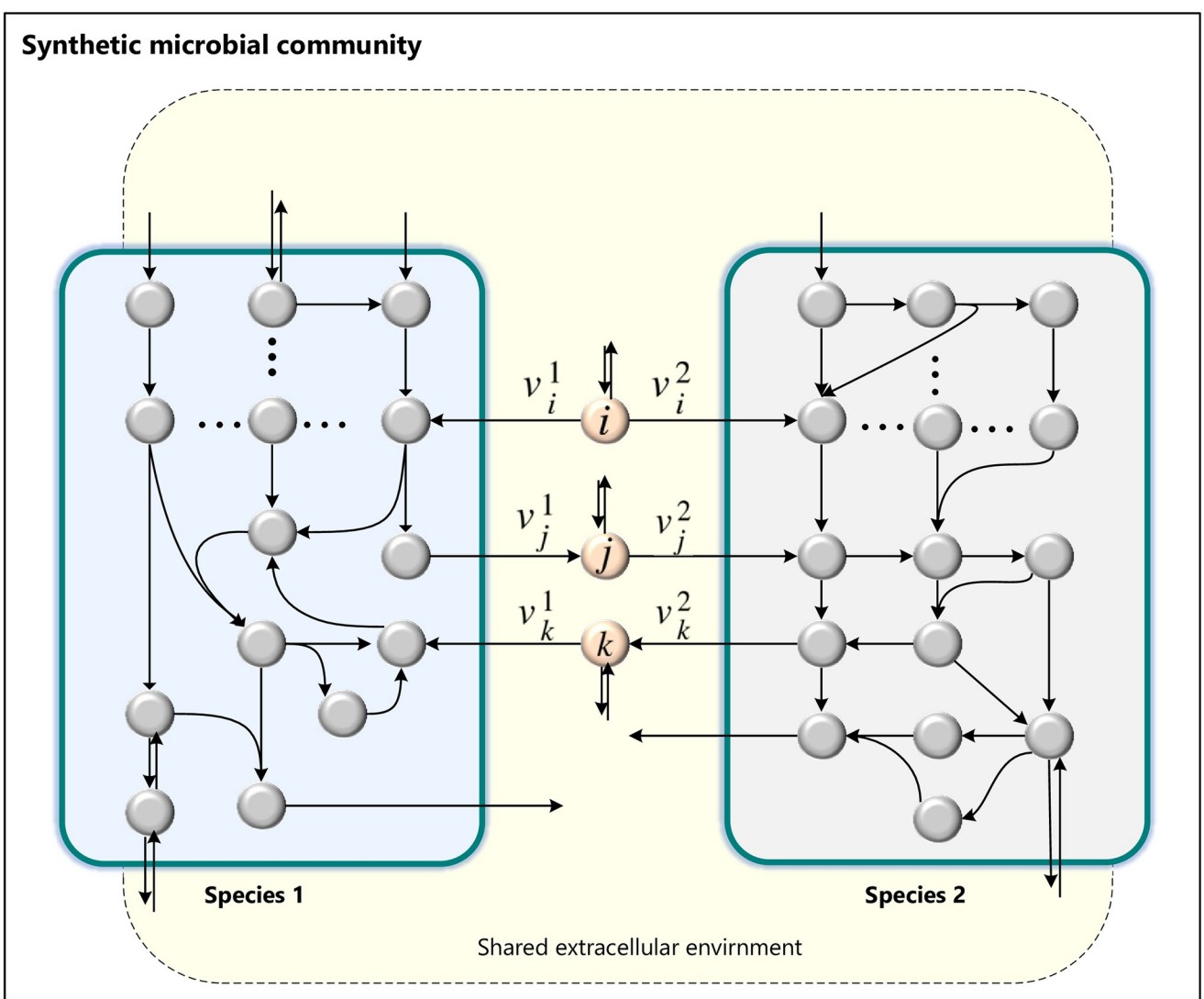

**Fig 1. Model for construction of a synthetic community.** A joint model for pairwise communities unites all species' GEM into one, with an extracellular environment for exchanging metabolites between the species.

### G.sulfurreducens and R.ferrireducens models

We also applied our framework to uranium-reducing and Fe(III)-reducing species, which are among the well-studied pairwise species. According to previous studies, *Geobacter sulfurreducens* and *Rhodoferax ferrireducens* compete for shared metabolites such as acetate, ammonium, and Fe (III) [42]. The availability of genome sequences provides the opportunity to reconstruct genome-scale metabolic models for these two microorganisms, which have been previously reconstructed [43].

### Preprocessing

First, we identify the blocked reactions and remove them, as done in some methods such as in [44]. Since we want to detect competed metabolites by deactivating them, therefore, if the

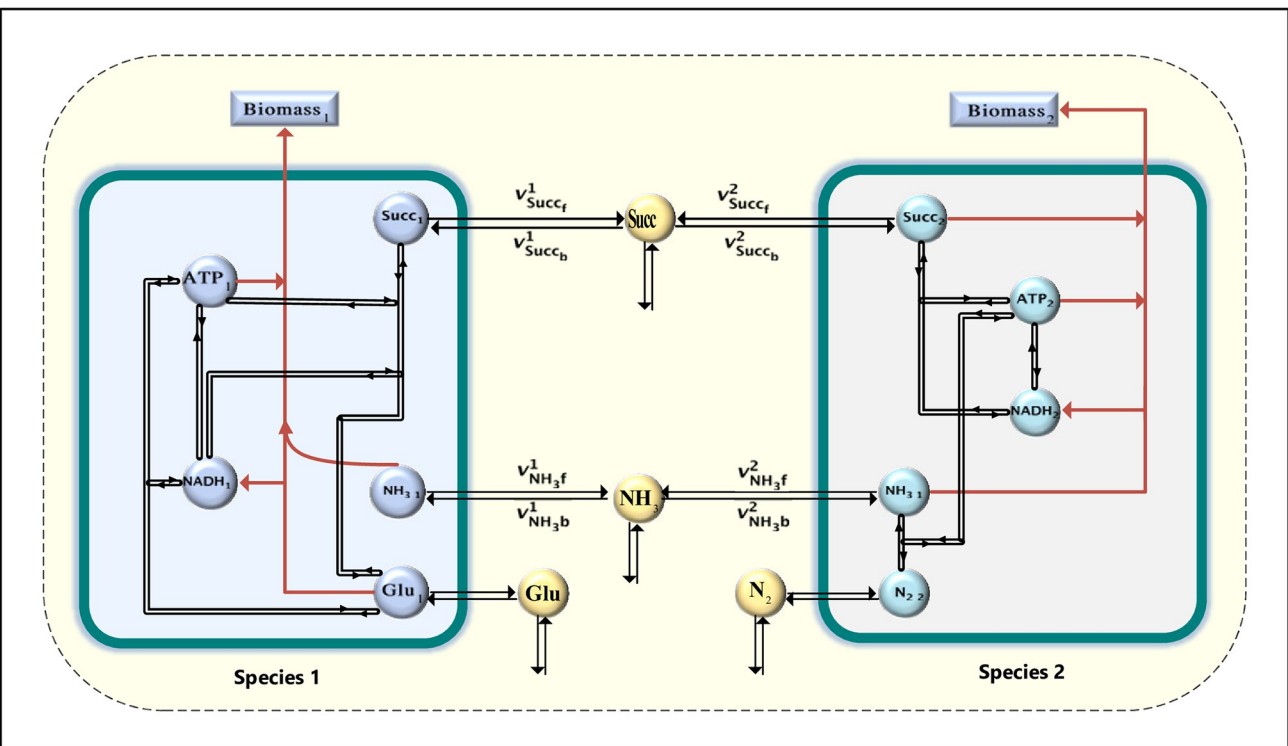

**Fig 2. The joint metabolic network model of two toy models.** Metabolites are represented with circles. Since all reactions in the toy model are reversible, we separate each reaction into two irreversible reactions.

reactions of the investigated shared metabolite are blocked reactions, the algorithm considers them as competed metabolites. Next, we decompose all reversible reactions into two separate irreversible reactions, namely forward and reverse reactions. We also need to compute the minimum flux of each exchange reaction for the computational model of commensalism/parasitism.

For these purpose, we used *flux variability analysis* (FVA) [44] where the flux of each reaction in the network is individually maximized and minimized while keeping the biomass flux fixed at a fraction of the optimal value obtained from *flux balance analysis* (FBA) [41]. We employed the *constraint-based reconstruction and analysis* (COBRA) toolbox [45], an open-source toolbox that contains methods (such as FVA and FBA) for the *in silico* study of genome-scale metabolic networks modeling.

## Computational model for competition

In constraint-based modeling of metabolic networks, the mass balance equation $Sv = 0$ is imposed by the assumption of a steady-state condition. Under this conditions, metabolic networks maintain a balance between the production and consumption of metabolites [46]. Additionally, irreversibility constraints can be introduced via the inequality $v_I \geq 0$, where $I \subseteq R$ denotes the set of irreversible reactions [47, 48].

We assume the steady-state flux cone of the community stoichiometric matrix $S^c$ as follows [49]:

$$C = \{v \in R^n | S^c v = 0, v_I \geq 0\} \tag{1}$$

where $S^c \in R^{m \times n}$ is the stoichiometric matrix of the synthetic community (Fig 1), which includes no blocked or reversible reactions. In our problem formulation, we represent a reversible reaction $v_i$ by two irreversible reactions, denoted as $v_{i-f}$ and $v_{i-b}$ (forward and backward), both with non-negative fluxes. As result, we have $I = R$. Here, we restrict our analysis to a subset $F \subset C$. We achieve this by constraining the fluxes to ensure that both species have at least a minimal growth rate and also requiring a non-zero exchange of matter with the environment. We refer to $v \in F$ as feasible flux distributions that allow both species to survive.

$$F = \{ v \in R^n | S^c v = 0, v^1_{biomaas} \geq \alpha . v^{1-max}_{biomaas}, v^2_{biomaas} \geq \alpha . v^{2-max}_{biomaas}, lb \leq v \leq ub, \exists i \in Exchange; v_i \neq 0 \} \, (2)$$

where $lb$ and $ub$ are the lower and upper bounds on the flux of reactions, respectively. $v^1_{biomaas}$ and $v^2_{biomaas}$ represent the flux of biomass reactions for organism 1 and organism 2, respectively. Additionally, $v^{1-max}_{biomaas}$ and $v^{2-max}_{biomaas}$ represent the maximum biomass flux for organism 1 and organism 2, respectively. In this formulation, we require the biomass flux in the network to be at least a pre-specified fraction ($0 < \alpha \leq 1$) of the maximum theoretical biomass flux identified beforehand through FBA.

Interactions among the microbes in a community can arise from competition for limited resources, which is mediated by metabolic exchange. Competition for metabolic resource can impact the growth or survival of one population in the presence of other species.

We consider competition interactions analogous to the concept of synthetic lethality [50], where only the simultaneous activation of two reactions $v^1_i$ and $v^2_i$ associated to common metabolite $i$ (as shown in Fig 1) is essential for the survival of both organisms. In the other words, the deactivation of both reactions ($v^1_i = 0, v^2_i = 0$) would lead to a violation of the constraints imposed on $F$ (survival). Competition interactions demonstrate that activating the flux of a common substrate reaction by one of the species can also result in the inactivation of the opposing microbe. This occurs because as one of the species consumes the shared metabolites, it leads to a decrease in metabolite concentrations and, consequently, a decrease in the growth of the other species. The survival of each of the species implies consuming the shared substrate, resulting in a decrease in the other organism's common substrate reaction flux. By proposed definition, even without prior knowledge of the community objective function, we can efficiently determine the competition for a common substrate in a metabolic network. Thus, we can consider an optimization feasibility problem, which is equivalent to having an objective function with an arbitrary constant value (such as zero) and a set of constraints defined in feasible set $F$.

We have developed COMMA (Competition Metabolic Assessment) as a mixed-integer linear programming algorithm to verify whether two species that share a common metabolite compete for its consumption. Assume that species 1 and species 2 consume a metabolite $i$ for their growth as shown in Fig 1. Note that this competition represents the trade-off between two microorganisms' uptake reactions $v^1_i$ and $v^2_i$, both of which utilize the same substrate. These two reactions compete for the consumption of the common metabolite. We define this issue as follows: an increase in the flux of $v^1_i$ leads to a decrease in metabolite concentrations for the other species, which in turn, results in a decrease in the growth rate of the opposing species or its the exclusion. An alternative definition is that activation of one of the two reactions involving limited shared metabolites leads to a steady state in the synthetic community only if the corresponding reaction of the opposing species carries zero flux ($v^1_i = 0 \Rightarrow v^2_i \neq 0$), where $v^1_i$ represents the flux of reaction for the shared metabolites $i$ in species 1. We have developed a COMMA method to simulate pairwise competition interactions involving common metabolites. The algorithm is applied to the joint stoichiometric model and does not require

the definition of the community's objective function. To determine whether two species compete for a shared metabolite, we verify the feasibility of the following program:

maximize 0

subject to:

$$v_i^1 = 0 \tag{3}$$

$$v_i^2 = 0 \tag{4}$$

$$v_{biomass}^1 \geq \alpha.v_{biomass}^{1\_max} \tag{5}$$

$$v_{biomass}^2 \geq \alpha.v_{biomass}^{2\_max} \tag{6}$$

$$S^c v = 0 \tag{7}$$

$$lb \leq v \leq ub \tag{8}$$

$$\theta_k.\beta \leq v_k, \qquad \forall k \in Exchange \tag{9}$$

$$\sum_{k \in Exchange} \theta_k \geq 1 \tag{10}$$

$$\theta_k \in \{0,1\}, v \in R^n \geq 0, 0 < \alpha, \beta \leq 1 \tag{11}$$

where $S^c$, $v_{biomass}^{1\_max}$, $v_{biomass}^{2\_max}$, $lb$ and $ub$ are defined as previously mentioned. Since all reactions are irreversible, the flux of all reactions are nonnegative. Eqs (7) and (8) are mass balance and flux capacity constraints. We ensured that each individual species has a minimal level of growth by adding Eqs (5) and (6). These two equations are constraints that ensure the biomass flux of each individual in the community is fixed to be at least a pre-specified fraction of maximum biomass flux (calculated by using FBA of COBRA toolbox). $\beta$ specifies the minimum flux value that at least one exchange reaction must have. In general, the values of $\alpha$ and $\beta$ depend on the specific application and can be chosen experimentally. In our implementation, based on relevant article [51], the values of $\beta$ and $\alpha$ are 0.01 and 0.1, respectively. $v_i^1$ and $v_i^2$ represent the flux of uptake reactions for shared metabolite $i$ in species 1 and species 2, respectively. *Exchange* is the set of exchange reactions of synthetic community and $\theta_k$ is binary variable that determines whether the flux of exchange reaction $k$ in the network must be non-zero. Eqs (9) and (10) indicate that algorithm searches and finds a medium in the space of flux distributions where both organisms have at least minimal growth rates without consumption of common metabolite $i$. If the program has no solution, this implies that there is no medium in which both species can survive without metabolite $i$. This means that the shared metabolite is essential for the growth of both strains, indicating that the two microorganism which share a substrate have the potential to compete for its consumption. The key idea of the algorithm is to analyze the subspace of flux distributions, taking into account not only the subspace of individual strains but also the flux space of the community and all the trade-offs and mutual interactions that two species can have for survival. The COMMA algorithm needs to be solved only once for each commonly consumed metabolite between the two species.

The idea behind our framework is that two species compete for a shared common substrate $i$ if the logical expression $v_i^1 = 0 \Leftrightarrow v_i^2 \neq 0$ holds true. Using the logical equivalent of this

expression, we have:

$$v_i^1 = 0 \Leftrightarrow v_i^2 \neq 0 \quad \equiv (v_i^1 = 0 \Rightarrow v_i^2 \neq 0) \wedge (v_i^2 \neq 0 \Rightarrow v_i^1 = 0)$$
$$\equiv (v_i^1 \neq 0 \vee v_i^2 \neq 0) \wedge (v_i^2 = 0 \vee v_i^1 = 0) \tag{12}$$

in this case, if the negation of the Eq (12) holds true, then we wouldn't have competitive interaction for the desired metabolite. The negation of the Eq (12) is:

$$\neg Eq \ (12) \equiv \underbrace{(v_i^1 = 0 \wedge v_i^2 = 0)}_{\text{I}} \vee \underbrace{(v_i^1 \neq 0 \wedge v_i^2 \neq 0)}_{\text{II}} \tag{13}$$

Consequently, we can say that if $(v_i^1 = 0 \wedge v_i^2 = 0)$, along with other constraints, hold true (feaasibility), then there is no competition for substrate $i$. We proposed an optimization algorithm for expression $v_i^1 = 0 \Rightarrow v_i^2 \neq 0$ (I). The expression (II) in Eq (13) indicates that if there exists a flux distribution where both species simultaneously consume metabolite $i$ and have pre-specified growth rates, then there is no competition. Therefore, in (II), we assume that the concentrations of metabolite $i$ in environments are sufficiently high. It is evident that when metabolite is abundantly available, two species will not compete. For this reason, we haven't solved any optimization algorithm for the case (II).

This strategy can be easily extended to triplets of microorgasisms or even larger groups by using the similar logical expression $(v_i^1 = 0 \wedge v_i^3 = 0 \Rightarrow v_i^2 \neq 0)$. In practice, we can run a mixed-integer linear programming (MILP) optimization with constraints such as $(v_i^1 = 0 \wedge v_i^2 = 0 \wedge v_i^3 = 0)$ to explore the competition.

## Computational model for commensalism/parasitism

In the community, microbes may have a positive impact on one species but a negative impact on another. Based on the definition provided in the article [19], in a parasitic relationship, species 1 could impose the production of metabolite $k$ on the species 2 and likewise, species 2 demands that the opposing species produce metabolite $j$, as shown in Fig 1. If the giver microbe supplies the metabolite while paying a fitness price, this interaction can be considered as parasitism. We developed CPARMA (Commensalism & Parasitism Metabolic Assessment), a linear programming algorithm based on this definition for parasitic interaction:

$$v_i^1 = v_j^{2(min\_flux)} \tag{14}$$

$$v_{biomass}^1 \geq (1 - \epsilon).v_{biomass}^{1-max} \tag{15}$$

$$v_{biomass}^2 \geq \alpha.v_{biomass}^{2-max} \tag{16}$$

$$S^c v = 0 \tag{17}$$

$$lb \leq v \leq ub \tag{18}$$

$$v \in R^n \geq 0, 0 < \alpha \leq 1 \tag{19}$$

where $S^c$, $v$, $v_{biomaas}^{1-max}$, $v_{biomaas}^{2-max}$, $\alpha$, $lb$ and $ub$ are defined as previously mentioned. $v_j^{2(min\_flux)}$ is the flux rate at which species 2 demands for metabolite $j$ from species 1 and represents the minimum flux rate of uptake reactions that species 2 requires for its growth. This value is calculated using the FVA function in the COBRA toolbox. $\epsilon$ is a positive number that is nearly zero. The

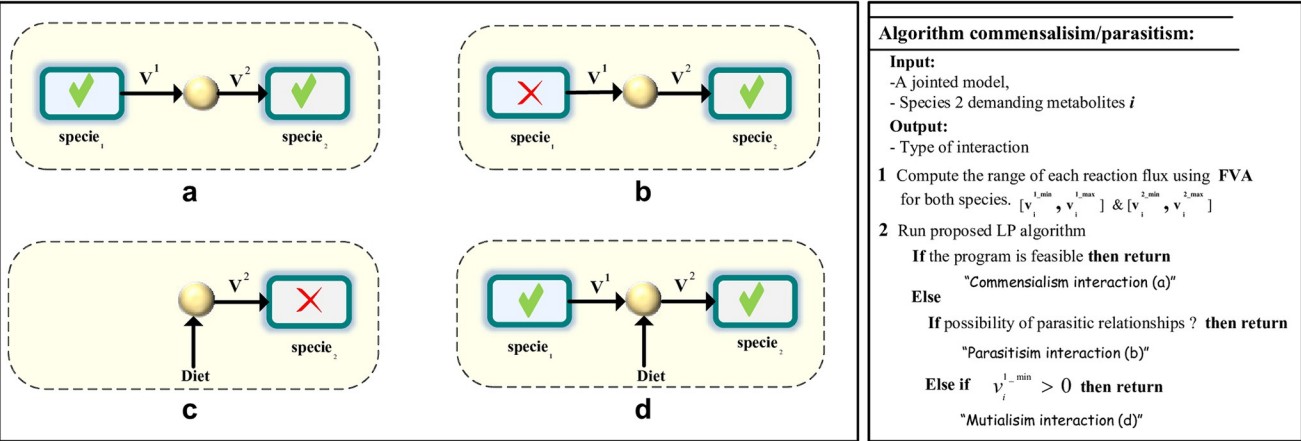

**Fig 3. The pseudocode and a graphical representation of the CPARMA algorithm outcome.** (a) Organism 1 can supply shared metabolites while its growth remains unaffected: This is known as commensalism interaction. (b) Species 2 is parasitic to species 1, and this relationship is termed parasitism. (c) Species 2 cannot survive in the absence of species 1 with just the concentration from the environment (d) Species 2 can survive by utilizing the supplied metabolites from both the environment and species 1.

user-specified threshold $\epsilon$ is used to determine the minimal biomass rate which species 1 must have while supplying metabolite $j$ to the species 2. In our computational model for commensalism and parasitism, the organism tends to counteract any externally imposed perturbations, such as producing opposing microbe need. This is achieved through the redirection of metabolic flux to have its own maximum biomass growth. It means that the giver microbe can provide the needs of other organism, while the organism still maintains maximum biomass production. For this reason, we choose 0.01 for the $\epsilon$. It suggests that if the microbe can couple growth production (organism fitness objective) with the production of opposing organism needs, the giver-consumer relation will be commensal, otherwise it can be a parasite.

Eq (14) indicates that species 2 imposes a demand on species 1 to produce at least the minimum required amount of metabolite $j$, and species 1 acts as the host to fulfill this demand. With this linear programming, we determine the feasibility of this demand and supply relationship.

If the program has a solution, then species 1 can provide metabolite $j$ to species 2 without paying a fitness price. In this case, microbe 2 can grow on the waste product of species 1, and we can consider this as a commensal interaction as shown in Fig 3(a). Otherwise, the demands from species 2 have a negative effect on the opposing microbe. We can consider this condition from two aspects. The first aspect is that species 2 can impose on species 1 to produce its demands, thus species 2 can be a parasitic to species 1, as described in Fig 3(b). The second aspect is that organism 1 can provide only a partial required amount to organism 2. If organism 2 can obtain the rest of its needs from the environment, this will lead to mutual interactions between the two organisms for this metabolite as shown in Fig 3(c) and 3(d). Due to the fact that this association is vital for species 2, according to [19], this type of relationship is called syntrophy. The pseudocode of the algorithm is shown in Fig 3.

We applied this algorithm to each shared metabolite where one species is the consumer and the other is the producer among the two organisms.

## Results and discussion

In microbial community biology, a major goal is to understand the foundation of microbial diversity. The study of microbe-microbe interactions plays a crucial role in addressing this

challenges. Methods for predicting interactions using GEM for each microorganism can be broadly divided into three groups: growth-based, abundance-based, and metabolite-based.

Growth-based methods predict interaction types by comparing the growth rates computed in the presence and absence of another species. Included in this group are the following methods: Klitgord et al. method [28], OptCom [19], The Microbiome Modeling Toolbox (MMT) [23], MICOM [5] and PyCom [52].

Abundance-based methods focus on the abundance ratio in co-culture versus mono-culture when deciding interaction types. An example of this group is COMETS [39].

The production or consumption of specific metabolites determines the type of microbe-microbe interaction and its effect on the abundance of community members [53]. There are some approaches, such as Metabolic Resource Overlap (MRO) [29] and ReMIND [39], that find a minimal medium for every member in the community species, and the common metabolites obtained from this minimal medium are used to compute the strength of competition interactions.

## Comparison to other methods

In Table 1, we summarize a comparison of these methods, including the strategies they use for prediction and the types of interactions they can predict. Additionally, we consider whether these tools are based on steady-state (static) or dynamic models. Dynamic models describe continuous changes in metabolic exchanges over time for the optimization problem. We also consider whether these methods provide quantitative metrics for deciding the types of interactions based on these metrics. As shown in Table 1, Metabolic Resource Overlap is the only method that proposes quantitative metrics for deciding competitive interactions.

One challenge with algorithms based on growth or abundance lies in their inability to distinguish between mutualistic and commensalistic interactions. Additionally, these algorithms cannot identify metabolites that lead to competition, commensalism, or other types of interactions, they predict the overall relationship. Among species, there can be competitive relationships concerning some metabolites, while for others, interactions may involve parasitic, mutualistic, or non-interactions. Additionally, some of these types of algorithms generally make an overall decision about the positivity or negativity of the relationship [34]. We compare our proposed algorithms with MRO and the competition score proposed by [54], which provides us with a comparable metric, as well as OptCom and MICOM, both of which are

**Table 1. Comparison of the COMMA and CPARMA methods with other interaction prediction tools using metabolic models.**

| Method | Interaction determinants | Steady-state/ Dynamic | Convex | Interaction type | Quantitative metrics |
|---|---|---|---|---|---|
| Klitgord et al. [28] | growth | steady-state | | commensalism/mutualism/neutral | |
| OptCom | growth | steady-state | | positive/negative | |
| MRO | metabolite | steady-state | | positive/competition | ✓ |
| MMT | growth | steady-state | ✓ | commensalism/mutualism competition/amensalism/parasitism/ neutral | |
| MICOM | growth | steady-state | ✓ | positive/negative | |
| COMETS | abundance | dynamic | ✓ | positive/negative | |
| PyCom | growth | steady-state | ✓ | cross-feed | |
| ReMIND | metabolite | steady-state | | competition/cross-feed | |
| COMMA | metabolite | steady-state | | competition | ✓ |
| CPARMA | metabolite | steady-state | ✓ | commensalism/parasitism/mutualism | ✓ |

well-known static methods. Additionally, we evaluate our algorithms using well-studied pairwise interactions and empirical studies.

## Toy model

To determine whether two organisms have the potential to compete for a shared substrate, we developed a mixed-integer computational metabolic modeling. To validate our framework, we first applied it to the simplified microbial pairwise introduced in [41]. As shown in Fig 2, both species can consume and produce the metabolites succinate and ammonia.

We solved the COMMA algorithm with $\alpha = 0.1$ and $\beta = 0.01$, first for the uptake reactions of the succinate metabolite, denoted as $v^1_{succ\_b}$ and $v^2_{succ\_b}$. Next, we also solved for the uptake reactions of the ammonia metabolite, represented by $v^1_{NH_3\_b}$ and, $v^2_{NH_3\_b}$. In this case, we considered that both species consume the shared metabolites but cannot produce them. The results indicate that these two species have the potential to compete for ammonia. According to the stoichiometric matrix of both species, it can be observed that they consume ammonia, but not succinate, for their biomass production. It's worth mentioning that our algorithm considers the two species to have no competition if both of them can grow at the considered minimal growth rate, rather than just one of them. This means that in our example, species 2 has alternative pathways to produce ammonia for its growth while species 1 does not have such pathways. We consider the restriction that both species have a pre-specified growth rate because there is a possibility that this pathway may be inactive and also both microorganisms can survive.

If we assume that the survival of one species does not cause competition, we could run two mixed-integer linear programming models: one with the deactivation of $v^1_i = 0$ along with constraints (5) to (11) from the COMMA computational model ($v^1_i = 0$ & (5–11)), and another by deactivating $v^2_i = 0$. When both programs in Eq (20) are infeasible, we can predict the possibility of competition.

$$[\nexists\, v;\ (v^1_i = 0 \,\&\, (5-11))] \wedge [\nexists\, v;\ (v^2_i = 0 \,\&\, (5-11))] \tag{20}$$

Therefore, the result of our algorithm predicts that the deactivation of $v^2_{NH_3\_b}$ leads to activation of $v^1_{NH_3\_b}$, and vice versa. However, both species would not compete for the succinate metabolite and, with a limited amount of succinate, they can have at least a minimal growth rate. It should be noted that the result of the algorithm also depends on the expected rate of growth we consider for the species ($\alpha$). We have also developed a computational framework to identify whether the exchange of metabolites between microbes is associated with a decrease in growth rate or not. Now, we assume two species have a giver-consumer relationship. We solved four CPARMA problems for the joint toy model: one pair of linear programming for succinate and another pair for ammonia. In each program, we assumed that one of the species in the pairwise community imposes on the opposing species to secrete the required metabolite. The results of this algorithm indicate that both of the species can supply each other's ammonia needs. We can consider this result as indicating that each one of the two species can grow on the waste product of the other species' succinate metabolite, which can indeed be considered a form of commensal interaction. But if each species imposes additional demands on the other to produce its required succinate, neither species can do it without paying a fitness cost. This result can be interpreted in two ways. In the first scenario, if a species has the ability to force others to produce the succinate metabolite, we can consider this interaction as parasitism. Otherwise, the opposing species can only partially provide for the needs. In such a case, the species will survive if the rest of the species' needs are provided by the environment.

## D.vulgaris and M.maripaludis models

The second pairwise community, we have used to verify our algorithm is *Desulfovibrio vulgaris* and *Methanococcus maripaludis* pair. Our COMMA algorithm predicted that there is no competition between the two species, which is consistent with [28], as these two species are giver and consumer. By applying our CPARMA method, we predict that for metabolites $H_2S$ and Formate, there is commensal interaction between the two species. This interaction is such that *M.maripaludis* can grow on the waste products of the *D.vulgaris*. However, for the ammonia and $H_2$, the algorithm couldn't find any feasible solution. This means that none of the species are able to produce the required metabolites of the other species without paying a fitness. In other words, the production yield of these metabolites is lower than their consumption. As previously mentioned (see Fig 3(c) and 3(d)) there is a possibility that the interaction is mutual. According to the results of our CPARMA algorithm, in any medium where ammonia and $H_2$ are exchanged between two spices, neither species can survive in the absence of the other. As in [28], the interaction in each medium, which is mutual, is due to the presence of ammonia and $H_2$.

## G.sulfurreducens and R.ferrireducens models

We used our framework to analyze the competition relationship between the pair of *G.sulfurreducens* and *R.ferrireducens*. Our competition algorithm predicted that these two species have the possibility of competing for the acetate, calcium, CO2, fumarate, potassium(k+), L-malate, magnesium, phosphate, sulfate and succinate. According to the studies by [42] and [43], these two species compete for acetate, ammonium and Fe(III). Acetate is included in the list of competed metabolites in our results. For ammonium and Fe(III), our algorithm predicts that the absence of acetate, fumarate and Fe(II) leads to competition. Since these species reduce Fe(III) to Fe(II) while oxidizing acetate [55], the outcome is not unexpected. Our algorithm predicts that both species can compete for phosphate and sulfate. This agrees with the result in [39]. Our results also indicate that, in addition to acetate, the competition for fumarate is equally important because in most alternative media, when shared metabolites are inactivated, it exists. The result of the CPARMA algorithm shows that the giver-consumer relationship between the two species is commensalism.

## Predicting metabolic interactions in gut microbiome and phyllosphere community

In this study, our COMMA and CPARMA approaches were adapted to investigate the microbial interactions within a honeybee gut microbiome. This community exhibits significant similarities to the human gut microbiome and is used as a microbiome model in a study by [39]. To quantify the competition of a given community for shared metabolites, we computed the number of shared metabolites for which our algorithm predicts the possibility of competition for their consumption, as well as the number of common metabolites that overlap between two species. The results of this analysis are summarized in the chord diagram found in Fig 4.

According to our results on overlapped metabolites, which are the common metabolites that two species may compete for their consumption, the highest metabolite overlap occurs between *L. kullabergensis* and *G. apico*, as well as between *L. kullabergensis* and *L. apis*, *G. apico* and *L. apis*, *L. mellifer* and *L. mellis*. The lowest metabolite overlap was found between *S. alvi* and *L. mellifer*. These findings align approximately with the results published in [39]. In contrast, the results of our analysis indicate that the highest competition rate is between *S. alvi* and *L. mellifer*, and the next competition rate is between *L. mellis* and *S. alvi*. The closeness of the

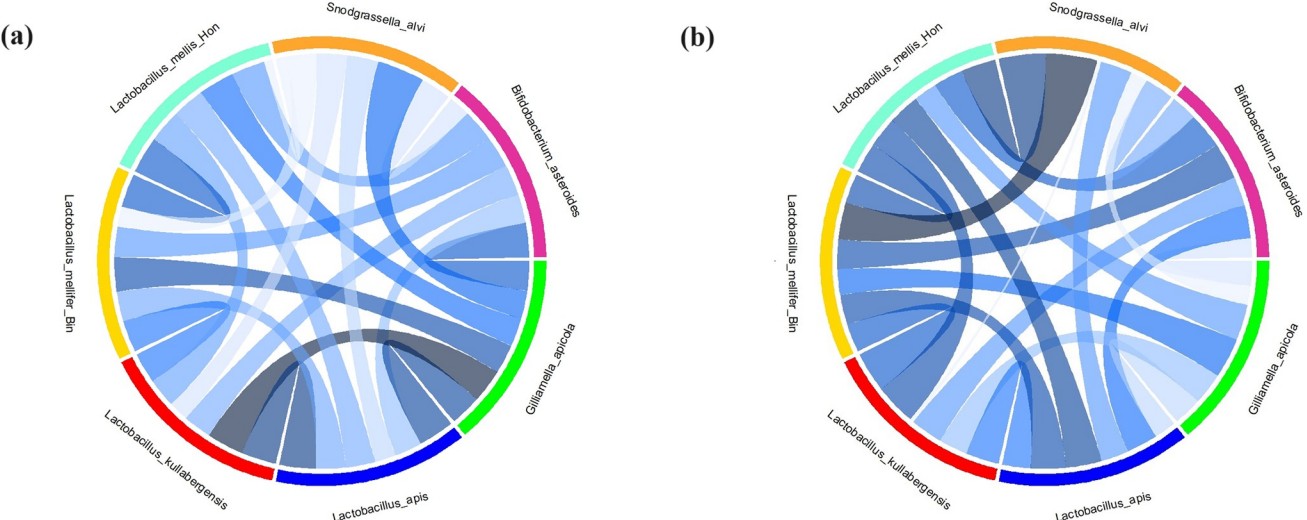

**Fig 4. Predicted competed and overlapping metabolites of pairwise interactions between seven microbial species.** (a) The number of overlapped metabolites. (b) The competition rate. Each color of the chord in the diagram is unique to one of the 21 pairs of species. The degree of color and widths of the chords increases from alice blue to dark blue as the competition rate and overlapped metabolites increase.

competition rates between these two species with *S. alvi* is excepted, based on the fact that the two species *L. mellis* and *L. mellifer* are closely related. However, for these two organisms, despite their dissimilarity and lowest resource overlap, they have the highest competition rate. Also, by comparing the interactions of *S. alvi* with other community members, it is apparent that *S. alvi* has the lowest competition rate with *G. apicola*, despite having the highest nutritional overlap with it. Our results provide insight into the fact that between two species that share a higher number of common metabolites for consumption, the possibility of alternative metabolites, rather than the competed metabolite, being utilized by species can be high. The commensalism relationships exist between all pairs of species except for the relationship between *L. mellifer* with the species *L. kullabergensis*, *L. mellis* and *L. alvi* and also the interaction between *L. mellis* and *S. alvi*. In the analysis of these pairwise giver-consumer relationships, the species cannot provide the needs for the survival of the opposing species without paying a fitness (loss of maximum growth rate).

In our final core analysis, we used seven phyllosphere bacterial strains that have been previously studied and cultured, as referenced in [38]. This article focused on the epiphyte strain Pe299R, in conjunction with six different phyllosphere-associated bacteria. We used genome-scale metabolic modeling of these species to predict the interactions between Pe299R species and other bacteria.

We used three tools: OptCom, MRO, and MICOM, to compare and verify our results. The MRO, which serves as an indicator of resource similarity, is the main competition metric in a related article by [38]. OptCom and MICOM are two flux balance analysis frameworks for predicting the growth rate of each participant in the community. These methods predict the bacterial competition based on growth rate differences and resource overlap between individual participants. MRO is a metric for predicting competition. For MICOM and OptCom, there is no specific quantitative criterion or threshold for deciding the type of interactions based on computed growth rate. Therefore, to quantify interactions, we used interaction strength,

**Table 2. Our algorithm competition score and dissimilarity metrics of phyllosphere-asscoiated strains in relation to Pe299R.**

| Strain | MRO | Phylogenetic distances | Competition score | OptCom (MOMA) | OptCom | MICOM | Our competition score |
|--------|-----|-----------------------|-------------------|---------------|--------|-------|----------------------|
| *PkP19E3* | 0.74 | 0.41 | 2.64 | 0.66 | 0.53 | 0.26 | 0.037 |
| *PssB728a* | 0.69 | 0.41 | 3.03 | 0.66 | 0.53 | 0.26 | 0.036 |
| *MethL85* | 0.67 | 0.68 | 0.92 | 0.63 | 0.5 | 0.25 | 0.038 |
| *SmFR1* | 0.69 | 0.62 | 1.05 | 0.64 | 0.5 | 0.25 | 0.059 |
| *ArthL145* | 0.77 | 0.76 | 2.41 | 0.7 | 0.5 | 0.25 | 0.04 |
| *RhodL225* | 0.62 | 0.76 | 0.69 | 0.5 | 0.03 | 0.27 | 0.047 |

which is calculated as the ratio of co-culture to monoculture growth rate:

$$Interaction\_Strength = \frac{v_{co}^{biomass}}{v_{mono}^{biomass}} \tag{21}$$

where $v^{biomass}$ represents the growth rate of an individual in monoculture (mono) or co-culture (co) [34]. When an interaction partner has a positive effect on a species, it leads to an interaction strength ratio above one. Conversely, if the effect is negative, the interaction strength becomes less than one [34]. We used OptCom firstly with the original strategy, as the inner problem's objective function involves the maximization of each species' biomass. Secondly, we employed the alternative function MOMA for the inner problem's objective function [34].

A competition score is computed based on the method developed by Chesson, as expressed by the following equation [54]:

$$Competition\_Score = \frac{\mu_i - 1}{\sqrt{\alpha_{ii}\alpha_{ij}}} \tag{22}$$

where $\mu_i$ is the species *i*'s growth rate in the monoculture, $\alpha_{ii}$ represents the competitive ability of species *i* when a near-isogenic strain of the same species *i* is present. This type of competition, known as intraspecies competition, occurs between individuals of the same species. Additionally, $\alpha_{ij}$ is competition coefficient of species *i* when a strain *j* is present. The results derived from our competition algorithm and other methods can be found in Table 2.

By applying the second algorithm to the six pairs, we found that there is no possibility for parasitic interaction between Pe299R and other strains. Since these bacteria are epiphytic bacteria, this result is not far from expected. Also, if there are any consumer and giver interactions between pairs, we consider this relationship as commensal (see Fig 3(a)).

MRO metric is used to assess the potential of species to compete for metabolic resources. Contrary to our findings, it indicates a high potential, as demonstrated in Table 2. Our algorithm's competition score does not correlate with the MRO result. Additionally, there is no correlation between the competition score of our algorithm and the competition score reported by [38] [r=-0.61, $\rho = 0.2$]. We also observed that the competition score of our algorithm is not correlated with phylogenetic distances, which is in agreement with [56, 57].

Since the interaction strength computed by OptCom and MICOM methods is less than 1, it confirms the negative nature of these interactions. This information represents microbial competition between pairwise species. According to the OptCom (original strategy) results, the highest competition interactions are between Pe299R and RhodL225, and the lowest are between Pe299R and PssB728a, as well as PkP19E3. This order is similar to the order of our COMMA algorithm output. But for MICOM, the lowest competition strength belongs to RhodL225. In general, all four approaches we have used for comparison have predicted competition between all these species and Pe299R.

**Fig 5. Population density of Pe299R over time.** The population size of Pe299R, in the presence of other strains, changes at different times. The total population density increased over time in a similar pattern as the population density in the monoculture also increased.

As indicated in Table 2, our results show that the potential for competition between Pe299R and the other species is low. Furthermore, the competitive abilities of the strains are completely opposite to the competition score of the related article [38]. Specifically, SmFR1 and RhodL225 and ArthL145 have the highest competition scores, while the lowest scores are for PssB728a and PkP19E3 and MethL85. This fact is also true for MRO value.

Microbial interactions can be inferred from changes in population size over time. Based on the biological perceptions, we can categorize the interactions between a pair of microbes as competitive if the population size decreases, and as cooperative if the population density increases [58].

Due to the difference in predictions, we aim to validate the outcomes of our algorithms through controlled empirical studies. We used an experimental dataset which was published in [38] to test whether we can validate our competition score with controlled empirical data. Our goal was to examine whether the experimental data supports competition between species. The relative increase in Pe299R population was estimated by CFU-based and single-cell reproductive success measurements. The data from this experiment is publicly accessible [38]. For the analysis of these data sets, we used statistical tests such as the Wilcoxon signed-rank test. As shown in Fig 5, although the related article represents these strains with a high resource overlap, which results in high competition among species, the population size of Pe299R increased over time in the presence of opposing bacteria. The p-values of the Wilcoxon signed-rank test were computed for CFU data of monoculture and in the presence of a second species. The results reveal that there is no statistically significant difference in the population density of Pe299R between monoculture and pairwise population. This indicates that the presence of the considered strains with the Pe299R does not have a significantly negative effect on its population size compared to monoculture. This suggests that low resource competition

explains the insignificant decrease in the population size of the Pe299R and our algorithm results are also consistent with this. The two-way ANOVA results from the related article also show similar outcomes. According to these analyses, we can infer that the potential level of competition interactions of an epiphytic species and Pe299R is not significant. The results are the same for the single-cell reproductive success experiment. There is no significant evidence of a difference in the reproductive success of Pe299R_CUSPER population compared to monoculture. Interestingly, the fraction of successful cells was higher in the presence of PssB728a and PkP19E3 and MethL85 [38]. Our pairwise competition scores demonstrate that the differences between scores are not significant [mean = 0.043, SD = 0.009]. Additionally, the competition score for Pe299R is lowest in the presence of PssB728a compared to other strains. To further verify our approach, we examined the metabolites for which there is a possibility of competition between MethL85 strain and Pe299R. The competition is only for the calcium, chloride, cobalt, copper, magnesium, manganese, zinc and potassium metabolites. This pair do not compete for the metabolite methanol and this result is in agreement with previous studies [38].

A significant advantage of the proposed approach is that there's no need for defining an objective function. Our algorithm is based on a qualitative analysis of reactions activities and therefore does not account for growth rate differences in the presence or absence of a species. We have investigated the interactions of a well-known mutualistic pair. The results show that for every metabolite for which our proposed CPARMA algorithm has no solution, if it does not appear in any medium, this causes mutual or commensal interactions. We observed these interactions for seven medium, as described in [28]. In both types of interactions, the opposing species couldn't survive in the absence of the giver species. Importantly, by considering the requirement of a microbe in the community to exchange matter with the opposing species, we are able to predict competition, commensal and mutual interactions. The competition between two microbes for a common metabolite represents the essentiality of two uptake reactions of common metabolites for each species. Because of this, our method can predict competition, and we can also distinguish between commensalism and parasitism, which the algorithms in [28] are not able to do. Despite the fact that many approaches compare the growth rates of pairwise species for interaction prediction, our results show no correlation with the estimated growth rate as mentioned in [38].

The metabolic resource overlap, defined as the fractions of shared metabolites in a minimal medium or set seed, is not effective in studying the competitive relationship between microbial pairs. For instance, we observed that Pe299R with other strains commonly consume more than a hundred metabolites. Specifically, PKP19E3 and PssB728a, which are closely-related species to Pe299R, have a higher population density compared to monoculture, despite having maximal resource overlap. Because of the low phylogenetic distance and, as a result, high metabolic similarity, the seed set detection algorithms that were used will include a large number of these metabolites, resulting in a high MRO value. This problem is especially relevant for the approaches which are graph-based. In our proposed methods, we consider steady state and mass balance. Furthermore, we verify that in the absence of common consumable metabolites, if there are alternative media for the surviving species, then there is no competition for the considered metabolites. We also consider other potential relationships between two species that could help for the survival of both species by exploring the feasibility space of the merged genome-scale metabolic network. Our approach is a similar strategy to that in [28], with the key difference being that we consider more than just minimal medium. Specifically, we search for alternative media in which two species can survive without consuming common metabolites. In conclusion, our algorithm's competition score provides insights for predicting

competition abilities in pairwise microbial communities and the relationship depends not only on resource availability but also on the potential of an organism to perform metabolic processes.

## Conclusion

The exchange of metabolites plays a crucial role in microbial community assembly. In this article, we developed methods to study metabolic interactions in a pairwise microbial community. We applied our framework to three well-studied pairwise interactions and two communities of gut microbiome and phyllosphere bacteria. We then compared our algorithm with MICOM, metabolic resource overlap (MRO), and OptCom. Our algorithm enables the analysis of interactions for each commonly exchanged metabolite and provides deeper insight to competition interactions. The results of the analysis can also serve as basis for designing a medium. We observed that there are pairwise species which are closely related and, as result, have more metabolic similarity but our results show a low potential for competition among them compared to other pairs. We discovered that our competition score can predict the outcomes of competition for both phyllosphere bacteria and the honeybee gut microbiome. For the honeybee gut microbiome community, the outcomes align with the results of previous studies, and it has been observed that in pairwise species with fewer nutritional overlaps, the competition rate was higher. The results are also consistent with the findings from the phyllosphere experiments. However, the resource overlap metric couldn't explain the Pe299R cell division and its population density. Overall, our results demonstrate that these interactions depend on the metabolic capacity of both species and predict interactions by examining the entire feasible flux space. Our proposed algorithms can serve as a valuable tool in the future for predicting essential pathways in the genetic engineering of microorganisms, with the goal of achieving maximum efficiency.

## Acknowledgments

This article is part of Soraya Mirzaei's Ph.D. program at Sharif University of Technology, co-supervised by Dr. Morteza Fotouhi and Dr. Mojtaba Tefagh. We are also grateful to Zainab Motasharei for her enlightening conversations.

## Author Contributions

**Conceptualization:** Soraya Mirzaei, Mojtaba Tefagh.

**Data curation:** Soraya Mirzaei.

**Formal analysis:** Soraya Mirzaei.

**Investigation:** Soraya Mirzaei, Mojtaba Tefagh.

**Methodology:** Soraya Mirzaei, Mojtaba Tefagh.

**Project administration:** Mojtaba Tefagh.

**Resources:** Soraya Mirzaei, Mojtaba Tefagh.

**Software:** Soraya Mirzaei.

**Supervision:** Mojtaba Tefagh.

**Validation:** Soraya Mirzaei, Mojtaba Tefagh.

**Visualization:** Soraya Mirzaei.

**Writing – original draft:** Soraya Mirzaei.

**Writing – review & editing:** Mojtaba Tefagh.

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
