## [Decision Letter · Decision Letter 0]

22 Mar 2024

Dear Dr. Tefagh,

Thank you very much for submitting your manuscript "GEM-based computational modeling for exploring metabolic interactions in a microbial community" for consideration at PLOS Computational Biology.

As with all papers reviewed by the journal, your manuscript was reviewed by members of the editorial board and by several independent reviewers. In light of the reviews (below this email), we would like to invite the resubmission of a significantly-revised version that takes into account the reviewers' comments.

We cannot make any decision about publication until we have seen the revised manuscript and your response to the reviewers' comments. Your revised manuscript is also likely to be sent to reviewers for further evaluation.

Sincerely,

Jie Li

Guest Editor

PLOS Computational Biology

Ruth Baker

Section Editor

PLOS Computational Biology

Reviewer's Responses to Questions

**Comments to the Authors:**

Reviewer #1: Soraya Mirzae et. al. have developed an algorithmic to predict the microbial interaction for co-culture systems. The authors have developed a graph-based algorithm, which does not require any biomass objective function and can predict microbial interaction under minimal growth conditions. The toy model of D. vulgaris and M. maripaludis has been used to validate the predictive ability of the algorithm by forming a pairwise community model. Possibilities of microbial competition, parasitism, commensalism, etc., have been explained by the pair of two species when they have shared common metabolites. Using the toy model, the authors have mentioned why metabolic uptake for species 2 is important to achieving the parasitic relation in association with species 1. To check the efficiency of the developed algorithm, they used the pairs of epiphyte strain Pantoea eucalypti (Pe229R) and different closely related phyllosphere bacterial strains. However, there are additional matters that require clarification in an updated version. Further detailed feedback is outlined below. However, there are few areas, which may be addressed by the authors for an overall improvement of their study.

1. Mixed-integer linear programming method has been used to simulate pairwise competition interactions for common metabolites. The problem statement not clear as the objective function for MILP is missing. The MILP code maybe provided for clarity.

2. Throughout the modeling work, authors have used α = 0.1 and β = 0.01 as the maximum biomass flux (for species 1) and minimum exchange flux (for species 2). It is very much unclear how these values were obtained.

3. From Fig 3, it has been seen that species 2 can survive in the absence of ammonia, as it has an alternative pathway to produce ammonia. Therefore, we can say that the developed method can also be helpful in predicting important pathways for coexisting species. In the era of genetic engineering, this might be a useful tool. Authors can think about that and elaborate their hypothesis to improve the usefulness of their tool.

4. Is there any natural co-occurrence for the selected microbial species (toy models and test models)? If yes, please discuss a few sentences in the introduction part. So that readers can understand and correlate the modeling work with the naturally existing microbial community.

5. How this method can be extended to community of three microbes or more.

Minor

1. In line number 60, the authors have written that ‘we develop genome-scale computational modeling’. There is a lack of clarity in the statement, as they have utilized the published GEM models from Schlechter et al., 2023 (Ref. 36), which is mentioned in the “Materials and methods” section (Line no. 74).

2. Authors have mentioned that ‘All reactions in these models are reversible..’ while discussing the details of the ‘Toy model’. However, it has been observed that only the exchange reactions are reversible (according to Fig 3). Either change the line or discuss it properly.

3. Species names should be in italics. Throughout the manuscript, these have been written in normal text like ‘Desulfovibrio vulgaris’ (In line number 96).

4. A sentence should not be started with “we…” that is in line number 74. Instead of writing ‘ref.36’, please mention proper referencing of the article, from where the metabolic networks have been obtained.

5. In lines number 158 and 159, a fraction of the sentence (‘according to the definition’) appeared twice.

6. In Table 1, ‘compertition score’ might be ‘Competition score’. Please check.

7. The clarity of the provided figures should be improved.

Reviewer #2: The interaction between microbial communities and their surrounding environment constitutes a subtle ecosystem. It is worth exploring how microorganisms interact, compete, and coexist with each other. This manuscript could be accepted if the following points are taken care of.

1. The linear programming algorithm established in microbial parasitism can consider mutually unfavorable influencing factors and the adverse environmental conditions faced together. If so, how is it reflected? If not, provide your reasons.

2. The meanings of the numbers 19 and 23 in formula (16) need to be described in the paper.

3. The content in the image is very blurry, and you need to effectively improve the clarity of the image. A small number of annotations should be added next to the image to help readers understand.

Reviewer #3: The paper presents a GEM-based computational modeling approach for exploring metabolic interactions in a microbial community. The authors developed a computational model for a synthetic microbial community that can predict possible metabolite interactions between species. The approach was validated using a toy model and a syntrophic co-culture of Desulfovibrio vulgaris and Methanococcus maripaludis. The approach was also applied to a real-world case study involving Pantoea eucalypti 299R and six different phyllosphere bacteria. The following concerns should be addressed:

1. the authors should consider comparing your approach with several existing state-of-the-art methods for studying metabolic interactions in microbial communities.

2. there is a lack of comparisons with widely-known baselines in the field.More real-world datasets should be used to validate your methods along the comparing with current methods.

3. The references introducing the backgrounds, particularly those pertaining to similar methods, appear to be somewhat outdated.

**Have the authors made all data and (if applicable) computational code underlying the findings in their manuscript fully available?**

Reviewer #1: **No: **

Reviewer #2: None

Reviewer #3: Yes

PLOS authors have the option to publish the peer review history of their article (what does this mean?). If published, this will include your full peer review and any attached files.

Reviewer #1: No

Reviewer #2: No

Reviewer #3: No
---

## [Editor Report · Decision Letter 1]

3 Jun 2024

Dear Dr. Tefagh,

We are pleased to inform you that your manuscript 'GEM-based computational modeling for exploring metabolic interactions in a microbial community' has been provisionally accepted for publication in PLOS Computational Biology.

Best regards,

Jie Li

Academic Editor

PLOS Computational Biology

Ruth Baker

Section Editor

PLOS Computational Biology

---

## [Editor Report · Acceptance letter]

16 Jun 2024

PCOMPBIOL-D-23-01863R1 

GEM-based computational modeling for exploring metabolic interactions in a microbial community

Dear Dr Tefagh,

I am pleased to inform you that your manuscript has been formally accepted for publication in PLOS Computational Biology. Your manuscript is now with our production department and you will be notified of the publication date in due course.

With kind regards,

Zsofia Freund
